# Retrospective, Non-Interventional, Multicenter Study on the Effectiveness and Safety of Intravesical Bacillus Calmette–Guerin in Patients with Non-Muscle-Invasive Bladder Cancer: Real-World Experience from Six Hospital Centers in Greece

**DOI:** 10.3390/curroncol32010018

**Published:** 2024-12-29

**Authors:** Panagiotis Angelopoulos, Titos Markopoulos, Lazaros Lazarou, Andreas Skolarikos, Panagiotis Stamatakos, Georgios I. Papadopoulos, Charalampos Fragkoulis, Konstantinos Ntoumas, Napoleon Moulavasilis, Panagiotis Levis, Dimitrios Papanikolaou, Ioannis Sokolakis, Konstantinos Hatzimouratidis, Charalampos Tzanetakos, Marina Psarra, George Mavridoglou, Konstantinos Skriapas, Dimitra Akrivou, Dimitrios Karagiannis, Christos Noutsos, Andreas Georgiou, Konstantinos Hastazeris, George Gourzoulidis, Dionysios Mitropoulos

**Affiliations:** 12nd Department of Urology, Sismanogleio Hospital, Medical School, National and Kapodistrian University of Athens, 15126 Athens, Greece; angelopoulospanag@gmail.com (P.A.); titosmark@gmail.com (T.M.); lazarou_laz@hotmail.com (L.L.); andskol@yahoo.com (A.S.); 2Department of Urology, General Hospital of Athens G. Gennimatas, 11527 Athens, Greece; pvstamatakos@gmail.com (P.S.); gipapadopoulos@yahoo.com (G.I.P.); harisfrag@yahoo.gr (C.F.); ntoumask@yahoo.com (K.N.); 31st Department of Urology, “Laiko” Gen. Hospital, Medical School, National and Kapodistrian University of Athens, 11527 Athens, Greece; napomoul@hotmail.com (N.M.); panagiotislevis@yahoo.com (P.L.); dmp@atenet.gr (D.M.); 42nd Urology Department of Medical School, Aristotle University of Thessaloniki, 54124 Thessaloniki, Greece; dimpapanikolaou18@gmail.com (D.P.); sokolakisi@hotmail.com (I.S.); promahon@otenet.gr (K.H.); 5Health Through Evidence, 17456 Athens, Greece; c.tzanetakos@hte.gr (C.T.); m.psarra@hte.gr (M.P.); 6Department of Accounting and Finance, School of Management, University of the Peloponnese, 24150 Kalamata, Greece; ge.mavridoglou@go.uop.gr; 7Department of Urology, General Hospital of Larissa, 41334 Larissa, Greece; kostas.skriapas@hotmail.com (K.S.); dim.s.akrivou@gmail.com (D.A.); kdimitrios@yahoo.com (D.K.); 8Department of Urology, General Hospital G. Hatzikosta, 45445 Ioannina, Greece; noutsos95@gmail.com (C.N.); hastaz@otenet.gr (K.H.)

**Keywords:** Bacillus Calmette–Guerin, non-muscle-invasive bladder, multicenter study, Greece

## Abstract

Background: While the clinical application of SII-ONCO-Bacillus Calmette–Guerin (BCG) for non-muscle-invasive bladder cancer (NMIBC) is well established in Greece, there is a lack of real-world data on its effectiveness and safety. This retrospective, observational, multicenter, chart-review study aims to provide real-life data on the effectiveness and safety of SII-ONCO-BCG in patients with intermediate- and high-risk NMIBC. Methods: From January 2016 to December 2023, medical records from six hospital centers were reviewed for adult patients with histologically confirmed stage Ta or T1 NMIBC (with or without carcinoma in situ [CIS]) who received at least one maintenance course of SII-ONCO-BCG after induction. Tumor recurrence and progression were monitored at scheduled time intervals. Primary outcomes included recurrence-free survival (RFS) and progression-free survival (PFS), while adverse events (AEs) constituted secondary outcomes. Results: A total of 162 patients receiving SII-ONCO-BCG were enrolled. Among all patients, 145 (89.5%) patients were men, 88 (54.3%) aged 70 years or older, 103 (63.6%) had T1, 43 (26.5%) Ta, and 21 (12.9%) concurrent CIS. The median follow-up duration was 28.9 months (range, 5–36) and the mean BCG intravesical instillation courses were 13.7 (range, 9–27). After 3-, 2-, and 1-year follow-up, RFS rates of 85.2% (95% CI, 79.7–90.7%), 85.8% (80.4–91.2%), and 87.0% (81.8–92.3%) were observed, respectively. The corresponding 3-, 2-, and 1-year PFS rates were 96.9% (94.2–99.6%), 96.9% (94.2–99.6%), and 97.5% (95.1–99.9%), respectively. During the whole follow-up period, 24 (14.8%) patients experienced at least one AE. Conclusions: This real-world study demonstrates that SII-ONCO-BCG is an effective and safe treatment for patients with intermediate- and high-risk NMIBC.

## 1. Introduction

Urinary bladder cancer is the most frequently occurring malignant cancer of the urinary tract and stands as the ninth most prevalent cancer worldwide. It is the 17th most common cancer among women, while among men, it holds the 7th position [1]. Over 70% of newly diagnosed bladder cancer cases are different types of non-muscle-invasive bladder cancer (NMIBC), which include noninvasive papillary carcinoma (stage Ta), carcinoma in situ (CIS), and tumors invading the submucosa (stage T1) [2]. Approximately 50% of new bladder cancer cases are low grade, noninvasive papillary tumors with a low risk (<5%) of progression, whereas high-grade T1 tumors have a much greater risk of progression [3,4]. Most bladder cancer patients present with superficial disease, which can be effectively treated through surgical resection. However, these patients remain at risk of tumor recurrence throughout their lives, with approximately 88% experiencing recurrence if they survive for 15 years [4,5].

Bacillus Calmette–Guerin (BCG) is a well-established adjuvant treatment used alongside transurethral resection (TUR) that significantly lowers the risk of recurrence and progression in NMIBC [4,6,7]. BCG immunotherapy is recommended for patients with high-grade Ta, any grade T1, and CIS lesions. It is also indicated for patients who have not responded to intravesical chemotherapy for low-grade, low-stage tumors [4,6,7]. The precise mechanism of action of BCG as an intravesical therapy remains unclear, but it is known to trigger local inflammatory reactions and an immune response. The attachment of live, attenuated BCG bacteria to the bladder mucosa, mainly facilitated by fibronectin, and their uptake by tumor cells play a key role in initiating the antitumor immune response. BCG acts as a nonspecific immunostimulant attacking tumor cells through the activation of antigen-presenting cell macrophages, T and B lymphocytes, natural killer cells, and the production of cytokine-like interleukins, interferon-γ, and tumor necrosis factor-a, which lead to tumor cell lysis [8].

Many prospective, randomized trials have validated BCG’s effectiveness in decreasing tumor recurrence, progression, and mortality [4,7]. Lamm et al. demonstrated a statistically significant reduction in disease progression and mortality rate following BCG immunotherapy [9].

Although response rates may differ depending on the entry criteria, the primary approach for managing superficial bladder cancer involves repeated BCG instillations following TUR of the tumor. These instillations are traditionally administered using the six-week induction schedule established by Morales in 1976, followed by long-term BCG maintenance. Though different maintenance schedules have been used, the one adopted by the US Southwest Oncology Group (SWOG) (induction plus 3-weekly instillations at 3, 6, 12, 18, 24, 30, and 36 months) has shown a significant reduction in tumor recurrence and an improvement in patients’ survival when compared with induction therapy alone [10]. The European Association of Urology (EAU) guidelines recommend BCG immunotherapy maintenance for one year in intermediate-risk patients and three years in high-risk patients [7].

More than seven BCG strains are commercially available for intravesical instillation, including ImmuCyst (Connaught strain), Oncotice (Tice strain), Pasteur strain, Immunobladder (Tokyo 172 strain), BCG-Medac (RIVM strain), SII-ONCO-BCG (Moscow I, Russian strain), and ImmunoBCG (Moreau RdJ strain) [11,12,13]. A meta-analysis conducted by the European Organization for Research and Treatment of Cancer (EORTC) found no major differences in efficacy among various BCG strains used for intravesical instillation [14,15]. However, conflicting results from various studies have made this topic controversial [16,17].

SII-ONCO-BCG is a live freeze-dried preparation derived from attenuated strain of Mycobacterium bovis (BCG Moscow I, Russian strain) and is commonly used in Greek clinical practice. The reconstituted product contains 1 − 19.2 × 10^8^ colony-forming units. Although its clinical benefits have been demonstrated [18] previously, up-to-date efficacy and safety data are always useful, as real-world studies complement clinical trial data, providing a more comprehensive understanding of how treatments perform in diverse patient populations and real-world clinical settings. Thus, the present retrospective, observational, multicenter, chart-review study was designed to generate real-life effectiveness and safety data of SII-ONCO-BCG in patients with intermediate- and high-risk NMIBC in Greece.

## 2. Materials and Methods

### 2.1. Study Design and Patient Population

In this retrospective, observational, multicenter, chart-review study, medical records from six hospital centers, i.e., Laiko General Hospital, Sismanoglio General Hospital, Papageorgiou General Hospital, Hatzikosta General Hospital, G. Gennimatas General Hospital, and the General Hospital of Larissa, were analyzed. Eligible patients were adults with histologically confirmed stage Ta, T1 NMIBC, or CIS after diagnostic transurethral resection of bladder tumor (TURBT) who received at least nine intravesical SII-ONCO-BCG (80 mg) treatment instillations (at least one maintenance course after the induction course) between January 2016 and December 2023. The exclusion criteria included prior BCG treatment, a history of solid organ transplantation, or muscle-invasive bladder cancer (MIBC). Patients with low EAU risk stratification, a duration of follow-up less than 3 months, fewer than nine intravesical BCG instillations, severe infection, or active tuberculosis were also excluded.

The relevant patients’ data were recorded based on the SWOG treatment protocol schedule for intravesical instillation (6-week induction, followed by three weekly maintenance instillations at 3, 6, 12, 18, 24, 30, and 36 months). The collected data were divided into three main categories following the SWOG treatment protocol: baseline data, induction (6-week) data, and maintenance data (3–36 months). The baseline demographic and disease characteristics of the patients included sex, age, smoking habits (such as current smoker, former smoker, and no smoker), comorbidities (such as respiratory disease, diabetes mellitus, and cardiovascular diseases), and biopsy/pathology report (type of bladder tumor [primary/recurrent], number of tumors [single/multifocal], T staging [Ta/T1], tumor size [diameter < 3 cm/≥3 cm], tumor grade [low/high], CIS [yes/no]). The induction and maintenance data mainly included the number of intravesical instillations, monitoring tests (i.e., cystoscopy, biopsy, cytology, ultrasound), recurrence and progression status, and safety status.

The data were collected using a structured study database provided to all participating centers. Collection was caried out anonymously, with no information retained that could link a patient’s code number to their identity. The study was conducted following the principles of the Declaration of Helsinki.

### 2.2. Outcome Assessment

The primary outcomes were recurrence-free survival (RFS) and progression-free survival (PFS). Recurrence was defined as any urothelial carcinoma (UC) reappearance within the urinary bladder after the initial TURBT, with an interval of less than 3 months being considered a residual tumor. Progression was defined as stage or grade advance (such as pathology report of Ta to T1-4 or T1 to T2-4). The primary outcomes of RFS and PFS were estimated as the time from the date of surgery (TURBT) to the recurrence and progression event after a 3-, 2-, and 1-year follow-up, respectively. The secondary outcome was the incidence of adverse events (AE) over 3 years, with the type and severity of AEs documented and categorized from grade 1 to 3 according to the Cleveland Clinic Approach of BCG Toxicity [19].

### 2.3. Statistical Analysis

Continuous variables were summarized as means and standard deviation (SD) and nominal variables were summarized as frequencies (n) and percentages (%) of the evaluable population. The primary outcomes, RFS and PFS, were analyzed using Kaplan–Meier (KM) survival analysis. The KM method was chosen due to its nonparametric nature, which makes it appropriate for analyzing time-to-event data without assuming a specific distribution. Confidence intervals (CIs) for survival probability were calculated and the KM survival curves were plotted. The safety of the BCG treatment was analyzed by the frequency, type, and severity of AEs occurring. All statistical analyses were conducted using the latest version of the statistical software package IBM SPSS 29.0.

## 3. Results

### 3.1. Patient Characteristics

A total of 162 patients receiving SII-ONCO-BCG 80 mg were enrolled in the study (Figure 1). The median follow-up duration was 28.9 months (range, 5–36). Among all patients, 145 (89.5%) patients were men with a mean age of 70 years (range, 44–92 years; 54.3% ≥ 70 years), of whom 59 (36.4%) were current smokers. The most frequent comorbidities were cardiovascular diseases (72 (44.4%) patients) and diabetes mellitus (15 [9.3%]). Pertaining to tumor characteristics, 103 (63.6%) patients had T1, 43 (26.5%) had Ta, 21 (12.9%) had concurrent CIS, and 124 (76.5%) had high-grade tumors. Among other available data, 72.7% of patients had a single tumor, 62.2% had tumors smaller than 3 cm, and 81.7% had primary tumors. Over a 3-year follow-up period, the mean documented BCG intravesical instillation courses were 13.7 (range, 9–27), and 76 (46.9%) patients had ≥ 15 intravesical instillations (i.e., more than 1 year). The baseline demographic and pathological characteristics of the patients are summarized in Table 1.

### 3.2. Primary Outcomes: Disease Recurrence and Progression

Disease recurrence occurred in 21 (13.0%), 23 (14.2%), and 24 (14.8%) of the 162 patients during the first, second, and third year of follow-up (Table 2). The corresponding 1-, 2-, and 3-year RFS rates were 87.0% (95% CI, 81.8–92.3%), 85.8% (80.4–91.2%), and 85.2%, (79,7–90.7%), respectively. A KM curve shows the probability of RFS over 3 years (Figure 2).

Pertaining to disease progression, four (2.5%), five (3.1%), and five (3.1%) patients experienced progression during the first, second, and third year of follow-up (Table 2). The corresponding 1-, 2-, and 3-year PFS rates were 97.5% (95.1–99.9%), 96.9% (94.2–99.6%), and 96.9% (94.2–99.6%), respectively. A KM curve shows the probability of PFS at the end of 36 months (Figure 3).

### 3.3. Secondary Outcome: Adverse Events

The analysis indicated that 24 of the 162 patients (14.8%) experienced at least one AE (32 episodes in total) during the whole follow-up period. The most common AEs reported were hematuria (11 episodes) and dysuria (7 episodes) (Table 3), whereas all AEs were grade 1 (50%) and grade 2 (50%). Neither a life-threatening episode nor sepsis was reported during the study period. Overall, intravesical SII-ONCO-BCG 80 mg therapy seems to be well tolerated by all patients.

## 4. Discussion

BCG has become the gold standard for adjuvant intravesical treatment in intermediate and high-risk NMIBC. Numerous studies have validated its effectiveness in preventing tumor recurrence, tumor progression, and mortality [20,21,22,23,24]. However, drug-related toxicity, patient compliance issues, and BCG shortages often hinder the completion of the prescribed treatment regimen, potentially affecting survival outcomes. In this direction, several studies have shown that lower recurrence and progression rates are associated with more cycles being administered [25,26,27]. Based on published studies, no one specific BCG strain among the available options has shown a superior profile in terms of RFS and PFS [14,24,28].

Although the clinical and safety profile of SII-ONCO-BCG has been previously demonstrated [18,29,30], up-to-date efficacy and safety data remain valuable. Real-world studies complement clinical trial data and provide a comprehensive understanding of a treatment’s performance across diverse patient populations and real-world clinical settings. This evidence is essential for informing healthcare decisions, improving patient outcomes, and shaping healthcare policies. To this end, the present study was conducted to add relevant evidence to the existing literature, and it is the first retrospective, non-interventional, multicenter, real-world study reporting safety and effectiveness data for patients with NMIBC receiving SII-ONCO-BCG in Greece.

Our real-world findings showed that the estimated 3-year RFS and PFS rates for SII-ONCO-BCG 80 mg were 85.2% and 96.9% respectively, with a mean number of BCG instillation sessions of 13.7 (range, 9–27). In terms of safety, 14.8% of patients experienced at least one AE during the 3-year follow-up period. The most common AEs reported were hematuria and dysuria, whereas all AEs were grade 1/2. There were no life-threatening occurrences or cases of sepsis documented during the study period, indicating that intravesical BCG therapy was well received/tolerated by all patients. These findings are very important and corroborate the fact that the mainstay of management for superficial bladder cancer remains that of repeated BCG instillations, following TUR of the tumor. In particular, SII-ONCO-BCG, a commonly used BCG strain in Greece, was found to be an effective and safe therapeutic option for patients with intermediate- and high-risk NMIBC in routine clinical practice.

A post-licensure, open-label, prospective, comparative study investigated the efficacy and safety of SII-ONCO-BCG in patients with NMIBC. More specifically, a study [18] conducted in India compared the efficacy and safety of 80 and 120 mg doses of SII-ONCO-BCG (Moscow I, Russian strain) in patients with intermediate- and high-risk NMIBC. The study reported results similar to ours, with an estimated 3-year RFS rate of 86.79% and 84.31% for the 120 and 80 mg groups, respectively, and an estimated 3-year PFS rate of 94.34% and 84.31% for the 120 and 80 mg groups, respectively [18]. Similar findings were also reported in a study conducted by Chen et al. [31] that retrospectively compared two strains of BCG (Oncotice [Tice strain] and ImmuCyst [Connaught strain]). Chen et al. found that patients were associated with a 3-year RFS rate of 80.6% for the Tice group and 78.5% for the Connaught group, whereas the 3-year PFS was 95.3% for the Tice group and 94.7% for the Connaught group. Another retrospective study performed in Australia exploring the efficacy of BCG therapy in patients with NMIBC indicated that the 1-year and 5-year disease-free survival rates were 72% and 41%, respectively, for patients receiving BCG treatment [32].

In terms of AEs, our study results were consistent with previously published findings that most AEs were grade 1/2 [31,33]. Previous reports have shown that AEs occurred in a similar proportion of patients, with an incidence ranging from 13.5% to 42% [33,34]. It is important to note that our AE incidence lay at the lower limit of this range.

Ongoing research focuses on optimizing BCG therapy regimens and exploring combination therapies with immune checkpoint inhibitors [35] to further enhance the effectiveness of intravesical BCG in managing NMIBC and reducing the disease recurrence risk. As evidence continues to grow, healthcare providers must stay updated on the latest clinical findings and recommendations regarding the use of BCG therapy in patients with NMIBC. This will enable them to offer the most appropriate and effective treatment options to their patients, ultimately improving long-term outcomes and quality of life for patients with NMIBC. Further to that, targeted diagnostic and prognostic methods with the discovery and application of molecular biomarkers are anticipated to receive serious attention and gain popularity [36]. Molecular biomarkers bring an opportunity to diagnose tumors earlier, easier, and with greater accuracy, and help select suitable treatments for those patients in need [36].

This study provides robust real-world effectiveness and safety data of SII-ONCO-BCG in NMIBC, derived from multiple specialized hospital centers in Greece. Our sample size, although not statistically determined, adequately reflects local clinical practice and provides valuable clinical insights associated with SII-ONCO-BCG use in Greece. One strength of our study is the exclusion of patients with fewer than nine intravesical BCG instillations. Though this resulted in a smaller patient cohort after the exclusion, we were able to draw solid conclusions from patients receiving intravesical BCG treatment. The present study also has several limitations. These are mainly related to its retrospective nature that is often associated with missing data, reliance on recall, lack of homogeneity, loss of follow-up, and lack of control groups. Despite these limitations, retrospective studies can be valuable in generating hypotheses, exploring associations, and informing the design of future prospective studies. Additionally, the lack of a control group makes it challenging to directly compare the efficacy and safety of SII-ONCO-BCG with other treatment options. However, this was out of the scope of the present study, which aimed to lay the groundwork for a better understanding of the safety and effectiveness of SII-ONCO-BCG in Greek patients with intermediate- and high-risk NMIBC. Further investigations of the SII-ONCO-BCG clinical profile in diverse patient populations and clinical settings, as well as of patient-reported outcomes and treatment perceptions in NMIBC, would be valuable for improving patients’ well-being.

## 5. Conclusions

Based on this real-world evidence study, SII-ONCO-BCG was found to be an effective and safe therapy for patients with intermediate- and high-risk NMIBC in routine clinical practice. The present study provides valuable clinical insights for the management of patients with NMIBC.

## Figures and Tables

**Figure 1 curroncol-32-00018-f001:**
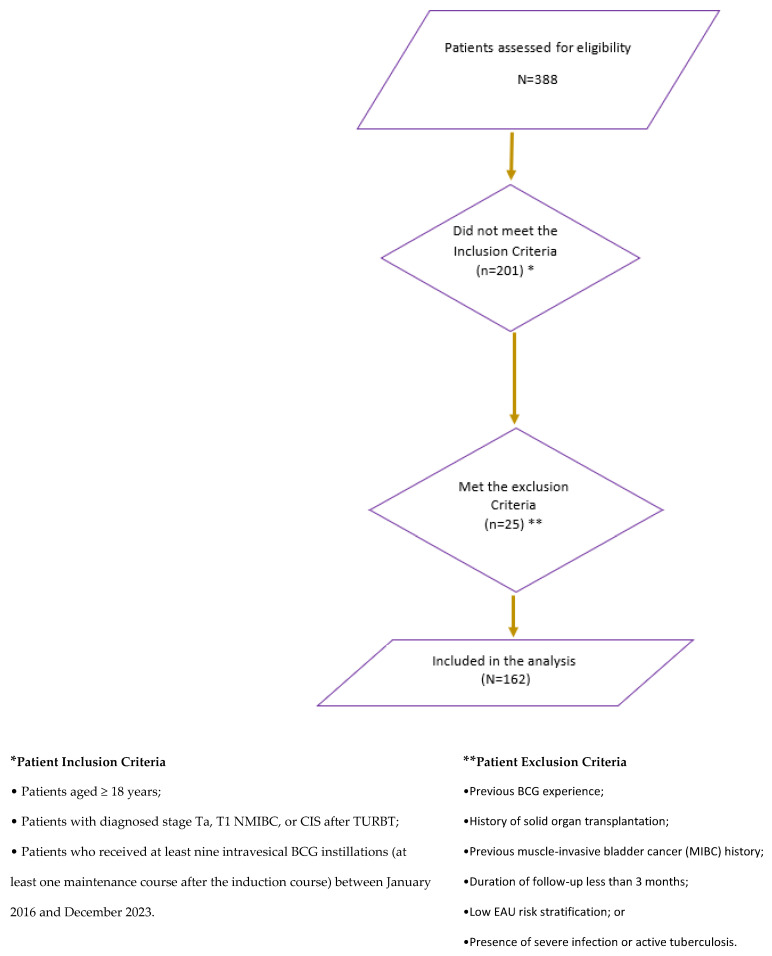
Flowchart of the study population.

**Figure 2 curroncol-32-00018-f002:**
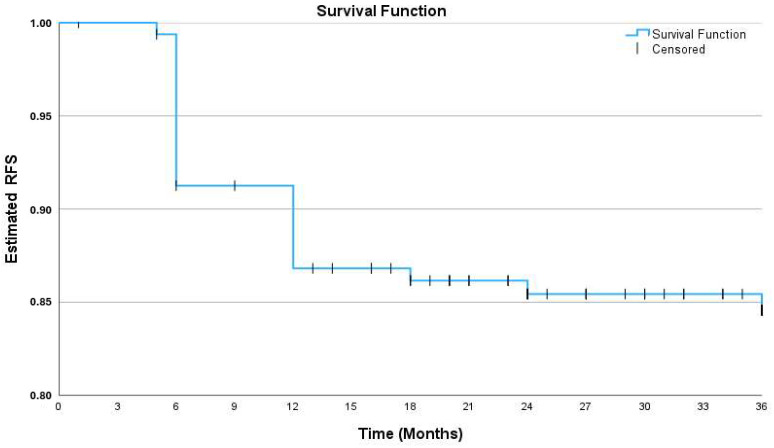
Kaplan–Meier survival curve of 3-year recurrence-free survival. RFS: recurrence-free survival.

**Figure 3 curroncol-32-00018-f003:**
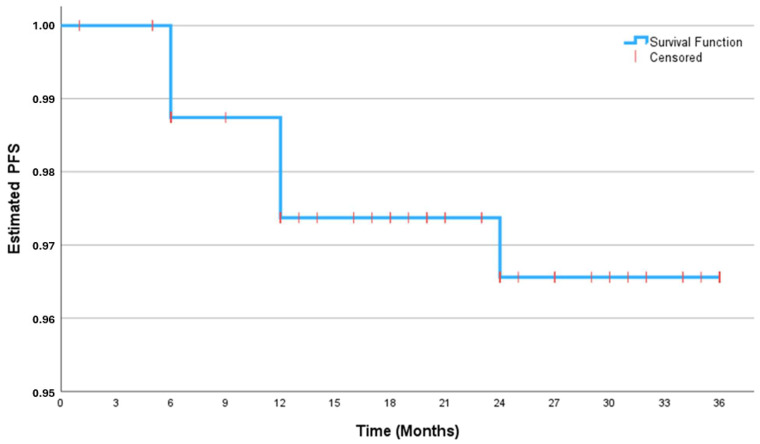
Kaplan–Meier survival curve of 3-year progression-free survival. PFS: progression-free survival.

**Table 1 curroncol-32-00018-t001:** Baseline patient characteristics.

Group Parameters	Total (*N* = 162)
**Age, mean ± SD (range), year**	70.0 ± 9.4 (44–92)
≥65 years, No. (%)	118 (72.8)
≥70 years, No. (%)	88 (54.3)
**Sex, n (%)**	
Male	145 (89.5)
Female	17 (10.5)
**Smoking n (%)**	
Smoking	59 (36.4)
Ex-Smoker	20 (12.3)
**Comorbidities n (%)**	
Cardiovascular diseases	72 (44.4)
Respiratory diseases	10 (6.2)
Diabetes mellitus	15 (9.3)
**Primary tumor n (%)**	*N* = 160
Primary	131 (81.7)
Recurrent	29 (18.3)
**Number of tumors n (%)**	*N* = 99
Single	72 (72.7)
Multifocal	27 (27.3)
**T category n (%)**	
T1	103 (63.6)
Ta	43 (26.5)
T1HG	89 (54.9)
**CIS n (%)**	
Pure	16 (9.9)
Concurrent	21 (12.9)
Any CIS	37 (22.8)
**Tumor diameter n (%)**	*N* = 90
<3 cm	56 (62.2)
≥3 cm	34 (37.8)
**Tumor grade n (%)**	
Low grade	38 (23.5)
High grade	124 (76.5)
**BCG treatment sessions**	
Intravesical instillation courses, mean ± SD (range)	13.7 ± 4.7 (9–27)
≥15 times of instillation n (%)	76 (46.9)

CIS: carcinoma in situ, BCG: Bacillus Calmette–Guerin, SD: standard deviation, cm: centimeter.

**Table 2 curroncol-32-00018-t002:** Survival analyses results.

Outcomes	Total (n = 162)
**Median follow up, months (range)**	28.9 (5–36)
**Recurrence-free survival analysis**	
1-year recurrence, No (%)	21 (13%)
1-year recurrence-free survival 95% CI	87% (81.8–92.3%)
2-year recurrence, No (%)	23 (14.2%)
2-year recurrence-free survival 95% CI	85.8% (80.4–91.2%)
3-year recurrence, No (%)	24 (14.8%)
3-year recurrence-free survival 95% CI	85.2% (79.7–90.7%)
**Progression-free survival analysis**	
1-year progression, No (%)	4 (2.5%)
1-year progression-free survival, 95% CI	97.5% (95.1–99.9%)
2-year progression, No (%)	5 (3.1%)
2-year progression-free survival, 95% CI	96.9% (94.2–99.6%)
3-year progression, No (%)	5 (3.1%)
3-year progression-free survival, 95% CI	96.9% (94.2–99.6%)

CI: Confidence interval.

**Table 3 curroncol-32-00018-t003:** Adverse event profile.

Adverse Events	No. of Patients with at Least One AE
Any AE, at Least One n (%)	24 (14.8%)
Dysuria	7
Fever	3
Hematuria	11
Increased frequency	4
Retention of urine	1
Urethral stricture	1
Urinary tract infection	5
Grade 1/2, n (%)	16 (50%)/16 (50%)

AE: Adverse event.

## Data Availability

The data that support the findings of this study are available from the corresponding author upon reasonable request.

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
