# Peer review of "Retrospective, Non-Interventional, Multicenter Study on the Effectiveness and Safety of Intravesical Bacillus Calmette–Guerin in Patients with Non-Muscle-Invasive Bladder Cancer: Real-World Experience from Six Hospital Centers in Greece"

_curroncol, 2024, doi:10.3390/curroncol32010018_

Round 1
Reviewer 1 Report (Previous Reviewer 2)
Comments and Suggestions for Authors
This second version of the paper is a great improvement, the authors are to be commended.
The manuscript has been much improved and is in a nice condition now.
Reviewer 2 Report (Previous Reviewer 1)
Comments and Suggestions for Authors
The authors did not fullfill all the comments. I Belize this study should be rejected
This manuscript is a resubmission of an earlier submission. The following is a list of the peer review reports and author responses from that submission.
Round 1
Reviewer 1 Report
Comments and Suggestions for Authors
This is an interesting study, however some major things should be considered while discussing this type of retrospective study.
1. Tha authors did not use the well established in literature and guidelines term "adequate BCG therapy". Instead they chose to use their own definition - it impacts future possibilities to compare this results to other studies.
2. In a retrospective cohort study like this it should be noted how many patients were screened in the study period, e.g. between 2016-2023 there were 300pts treated with BCG in six centers, out of which, 40 did not finish the induction, 15 were diagnosed with recurrence before maintenance, 10 refused maintenance and (...) eventually 180 were enrolled.
3. The number of cases is low for such a big multicetner study! For a NMIBC report 162 patients treated in 6 center through a period of 8 years means its a mean of 3,4pts/center/year! Bladder cancer should be treated in a specialized centers, and treating 3 patients per year is surely not enough.
4. There are A LOT OF MISSING DATA. eg: primary vs recurrent tumor was assessed in 60/162 patients!!!!, number of tumors 99/162!!!!, tumor size 90/162!!!! These are basic medical history information, and if this cannot be found in most of the files how can we trust other information.
Reviewer 2 Report
Comments and Suggestions for Authors
Summary of the Review
Study Design and Manuscript Contents
・The study is well-structured as a retrospective, multicenter observational chart review, focusing on a relevant clinical question regarding the effectiveness and safety of SII-ONCO-BCG in treating NMIBC.
・The research aims to contribute valuable real-world data that complements existing clinical trials, filling a significant gap in the literature.
Relevant Comments: Major Strengths and Major Weaknesses
Major Strengths:
・The study encompasses multiple centers, enhancing the generalizability of the findings.
・A clearly defined patient population with robust inclusion and exclusion criteria strengthens the study's internal validity.
・Comprehensive reporting of primary and secondary outcomes provides a detailed analysis of both efficacy and safety.
Major Weaknesses:
・The retrospective design inherently carries risks of selection bias and missing data.
・Lack of a control group limits the ability to compare the efficacy and safety of SII-ONCO-BCG against other treatments.
Minor Weaknesses
・The manuscript would benefit from a more concise writing style in certain sections to enhance clarity and readability.
・A clearer distinction between baseline and outcome data could improve the flow of information.
Detailed Comments
Title
1. Appropriate.
Abstract
2. Consider rephrasing the last sentence to emphasize the clinical implications of the findings.
Introduction
3. The introduction provides a solid background, but it could better integrate recent literature on BCG therapy to enhance the context.
4. Clarifying the rationale behind studying SII-ONCO-BCG specifically could strengthen the argument for its significance.
Materials and Methods
5. Ensure clarity in the statistical methods section; specifying the exact tests and software information in more detail would enhance reproducibility.
Results
6. The results section is detailed, but incorporating more graphical representations could improve understanding.
Discussion
7. The discussion effectively contextualizes findings within existing literature; however, more emphasis on how these results might influence clinical practice would be beneficial.
8. Consider discussing the current diagnostic aspects. For example, you might refer to PMID: 29755006 and PMID: 35933266 from the EAU guidelines 2024. You don't have to use these articles for discussion but should find relevant articles to enhance your paper.
Conclusion
9. The conclusion reiterates key findings but could better emphasize future research directions or clinical recommendations based on the results.
10. Consider framing the conclusion in a way that invites further exploration of SII-ONCO-BCG in diverse patient populations.
References
11. Ensure all citations are up-to-date, particularly concerning recent studies on NMIBC and BCG therapy.
Figures and Tables
12. Tables are well-organized, but including legends that explain abbreviations or clinical significance of presented data would enhance clarity.
